

# Pipes to Earth's subsurface: The role of atmospheric conditions in controlling air transport through boreholes and shafts

Elad Levintal[1], Nadav G Lensky[2], Amit Mushkin[3], Noam Weisbrod[1]

[1]Environmental Hydrology and Microbiology, The Zuckerberg Institute for Water Research, The Jacob Blaustein Institutes for Desert Research, Ben-Gurion University of the Negev, Midreshet Ben-Gurion 8499000, Israel.

[2]Geological Survey of Israel, Jerusalem 9550161, Israel.

[3]Department of Earth & Space Sciences, University of Washington, Seattle WA 98105, USA.

*Corresponding author:* Noam Weisbrod (weisbrod@bgu.ac.il)

**Abstract.** Understanding air exchange dynamics between underground cavities (e.g., caves, mines, boreholes, etc.) and the atmosphere is significant for the exploration of gas transport across the Earth-atmosphere interface. Here, we investigated the role of atmospheric conditions in air transport inside boreholes through in-situ observations. Three geometries were explored: (1) a narrow and deep *shaft* (0.1 m and 27 m, respectively), ending in a large underground cavity; (2) the same *shaft* after the pipe was lowered and separated from the cavity; and (3) a deep *large-diameter borehole* (3.4 m and 59 m, respectively). Absolute humidity was found to be a reliable proxy for distinguishing between atmospheric and cavity air masses and thus to explore air transport through the three geometries. Airflow directions in the first two narrow-diameter geometries were found to be driven by changes in barometric pressure whereas airflow in the large-diameter geometry was correlated primarily to the diurnal cycles of ambient atmospheric temperature. High $CO_2$ concentrations (~2000 ppm) were found in all three geometries, which can indicate that the airflow to the atmosphere is also significant to the investigation of greenhouse gas emissions.

## 1 Introduction

Understanding air transport between the Earth's subsurface and the atmosphere is a key element in the study of surface and near-surface biological, chemical, and physical processes. Air transport between the Earth and the atmosphere is commonly driven by diffusive and advective mechanisms (Choi and Smith, 2005; Ganot et al., 2014; Hillel, 1998; Kawamoto et al., 2006; Kuang et al., 2013; Noronha et al., 2017; Sánchez-Cañete et al., 2013) and has been mainly studied within soils (e.g., Allaire et al., 2009; Bayer et al., 2017; Choi and Smith, 2005; Zeng et al., 2017). However, as different types of natural or artificial boreholes and shafts also exist (e.g., Berthold and Börner, 2008; Kang et al., 2014, 2015), understanding the





mechanisms that govern air and greenhouse gas (GHG) transport in such conduits is important (Berthold and Börner, 2008; Perrier et al., 2005).

Boreholes and shafts are abundant discontinuities crossing the Earth's surface. Typically, these features can function as conduits between the ambient atmosphere and underground cavities (e.g., James et al., 2015; Moore et al., 2011; Pla et al.,
2016; Steinitz and Piatibratova, 2010). These underground cavities can be more than one-order of magnitude larger than the connecting conduit. Advective air transport in such boreholes or shafts can be governed mainly by: (1) barometric pumping (BP), which is the inflow and outflow motion of subsurface air due to pressure gradients governed by changes in barometric pressure (Ellerd et al., 1999; Neeper, 2002; Neeper and Stauffer, 2012; Perina, 2014; Perrier and Le Mouël, 2016; Rossabi and Falta, 2002; Thorstenson et al., 1998); and (2) thermal-induced convection (TIC), which develops when there are
unstable density gradients resulting from temperature differences between the atmospheric air and the borehole or underground cavity (Ganot et al., 2012; Nachshon et al., 2008; Perrier et al., 2002; Weisbrod et al., 2009; Weisbrod and Dragila, 2006).

BP will initiate airflow when pore pressure in the surrounding rock/soil differs from the pressure within the borehole or the underground cavity, which is considered as equal to atmospheric pressure (Kuang et al., 2013; Neeper, 2003; You et al.,
2010). BP is dependent on the surrounding rock/soil depth and permeability and on the barometric pressure changes (Massmann et al., 2000). A water table at the lower boundary is considered as impermeable for BP (You et al., 2010).

Temperature differences within the borehole or shaft and between them and the atmosphere above will determine the onset of TIC. Temperature differences can differ between locations and depend on several parameters, such as the surrounding rock/soil thermal properties, the geometry of the borehole, or the atmosphere temperature cycles (e.g., Eppelbaum and
Kutasov, 2011; Klepikova et al., 2011; You and Zhan, 2012).

Each of the above advective mechanisms was studied individually. However, to the best of our knowledge, no comparative research has been done on the dominating mechanisms between different borehole and shaft geometries, e.g., different borehole diameters. Therefore, the relative contribution of each mechanism to the overall air transport within boreholes or shafts under different environmental conditions remains loosely constrained.

Here, we investigated the role of atmospheric conditions on the air transport mechanisms inside three borehole geometries: (1) a narrow-diameter shaft (0.1 m) with a PVC pipe, opening into a large underground cavity (defined hereafter as "*shaft*" geometry); (2) the same shaft after the pipe was lowered and the link between the shaft and the underground cavity was blocked ("*borehole*" geometry); and (3) a borehole with a larger diameter of 3.4 m ("*large-diameter borehole*" geometry). Specifically, we aimed to assess the air inflow and outflow events by quantifying oscillation of physical parameters, mainly
temperature and relative humidity (RH) along the boreholes, and relate the flow events to the atmospheric forcing conditions.



## 2 Materials and Methods

We examined two sites: (1) A narrow-diameter (0.1 m) 27-m-deep borehole that reaches a large underground cavity located above the local water table of the southern part of the Israeli Coastal Aquifer, and (2) a large-diameter (3.4 m) 59-m-deep borehole that reaches the local aquifer near the Nabatean archeological site of Avdat in the Negev highlands of southern Israel.

The first borehole was drilled into a sequence of alternating layers of sand, sandstone and silt, comprising the unsaturated zone of the Israeli coastal aquifer (Goren et al., 2014; Schwarz et al., 2016). A PVC pipe (case), was inserted into the narrow-diameter borehole to prevent potential soil collapse. The pipe reached the top of the underground cavity, which is at least two orders of magnitudes larger in volume than the pipe and is located well above the local groundwater level of ~-80 m below the ground. The measurements lasted 46 days during the spring/summer of 2016 with only one rainy day during that period (13/4/2016). During the first 42 days of measurement (5/4/2016-16/5/2016), the underground cavity was connected to the atmosphere only by the PVC pipe ("*shaft*" geometry). Then, for an additional four days, the PVC pipe was lowered such that the end of the pipe touched the cavity floor, effectively disconnecting the underground cavity from the borehole and the atmosphere ("*borehole*" geometry). Therefore, we could distinguish between airflow when: (1) the borehole connects between the ambient atmosphere and the deep cavity, and (2) the borehole is disconnected from an underground reservoir and only connected to the ambient atmosphere.

The second borehole site ("*large-diameter borehole*") was excavated into Eocene chalk formations with an upper part of loess soil (Nativ et al., 2003; Shentsis et al., 1999). The water table in this site is at depth of ~55.5 m with small seasonal changes of less than 1 m, thus the 59 m deep borehole reached the local water table. The borehole did not have casings except for a few meters in the upper parts of the loess soil, and it was open to the atmosphere. The measurements at this site lasted for one week (20/2/2017-26/2/2017). The three geometries are illustrated in Fig. 1a.

Sensors installed at the first site included four thermocouples (type T, Omega Engineering, UK) at depths of: 0, 6, 18, and 24 m and two RH-temperature sensors (Hygroclip 2, Rotronic AG, Switzerland) at a depth of 12 m and at the lower part of the borehole at its connection point to the cavity (27 m depth); see Fig. 1b for an example of sensor locations. Outside the borehole, a meteorological station was installed, at 2 m above ground, including the following sensors: (1) wind velocity and direction (WindSonic, Gill Instruments, UK); (2) Barometric pressure (CS106, Vaisala, Finland); and (3) RH-temperature (same type as within the borehole). Data from all sensors were measured at 5-second intervals and averaged and logged at 10-min intervals (CR1000, Campbell Scientific, UT, USA). In addition, a televiewer was lowered into the pipe to verify that the pipe was intact and was connected to or disconnected from the underground cavity in the *shaft* or *borehole*, respectively. Measurements at the second site included a similar RH-temperature sensor at the depth of 10 m and a temperature sensor at 59 m.



Absolute humidity (AH) was used as a tracer for the air transport within the three geometries and was calculated from the measured temperature and RH data using Eq. (1). (Hall et al., 2016):

$$AH = \frac{6.112 \times e^{[(17.67T)/(T+243.5)]} \times 2.1674RH}{(273.15+T)} \tag{1}$$

where AH is in g/m$^3$, $T$ is temperature in $^\circ$C, and RH is in %. The use of AH as a tracer was previously reported by Neeper

5    (2003).

## 3 Results and Discussion

### 3.1 *Shaft* geometry

An example of one week time series results from the *shaft* is shown in Fig. 2a. Atmospheric temperature and RH presented typical daily cycles, as expected. During daytime, atmosphere air temperatures were higher with lower RH values (25-30 °C,

20-50 %, respectively) compared to night time (10-15 °C, 80-100 % respectively) (Fig. 2a -1 and 2, black lines). In contrast, air temperature and RH changes inside the *shaft* did not follow the daily cycle (Fig. 2a -1 and 2, purple and green lines). Measured temperatures at 12 and 27 m were 23.7 ± 0.7 and 24.7 ± 0.6 °C, respectively, with similar amplitudes. The RH sensors showed similar values at 12 and 27 m (Fig. 2a -2, purple and green lines, respectively): From the overall of 6050 measurements (42 days) at 12 and 27 m, 75 % of the RH values were above 95 %, and the remainder 25 % were no lower

than a minimum of ~50 % RH (Fig. S1 – supporting information).

Barometric pressure typically varied with two diurnal cycles; the average barometric pressure was 1008.68 ± 3.2 mbar (Fig. S1) with a rate of changes ranging from -4×10$^{-4}$ to 4×10$^{-4}$ mbar/min (Fig. 2a -4). For the majority of time (77 %) the temperatures at the lower part of the *shaft* (depths of 12 and 27 m) were higher than those measured in the atmospheric air with an average temperature difference of 6.1 ± 3.7 °C (Fig. 2a -5). Wind velocity at 2 m above ground was mostly calm (1-2

m/s) with daily peaks of 7-10 m/s in the afternoon, corresponding to the Mediterranean Sea Breeze (hours 13:00-18:00, Fig. 2a -6) (Lensky and Dayan, 2012).

Atmospheric AH was stable during the measurement period and values ranged between 10-15 g/m$^3$. Contrastingly, the AH at the underground cavity boundary was considered as a constant value of 22.7 g/m$^3$ according to T = 24.7 °C (the measured temperature at the *shaft*-cavity interface) and RH = 100 % (representing saturation conditions as suggested for underground

cavities by Bourges et al., 2014 and Perrier et al., 2005). AH values inside the *shaft* fluctuated between these two end values (cavity at the bottom and atmosphere at the top). Most of the time *shaft* AH values, 12 and 27 m, reflected cavity values (>20 g/m$^3$) and ~10 % of the time *shaft* AH values reflected atmospheric values, below 15 g/m$^3$ (Fig. 2a -3, purple and green lines). No marked difference was observed between the 12 and 27 m values. Considering the *shaft*'s impermeable perimeter



(i.e., the PCV pipe), AH changes in the *shaft* necessarily indicate inflow/outflow from the *shaft*'s lower and upper boundaries (i.e., the atmosphere or cavity air). We therefore regard low and high AH values in the *shaft* as indicators for down-welling mass flow (inflow) and upwelling mass flow (outflow), respectively.

To quantitatively define each inflow or outflow event, a classification algorithm was built and solved for the 42-days data using MATLAB$^{TM}$ software. The 12 m AH vector was transformed to a *dAH/dt* vector and then two conditions were defined as "must" so that an event is classified as an "inflow" or an "outflow" event: (1) *dAH/dt* < threshold value for an inflow event or *dAH/dt* > threshold value for an outflow event; and (2) the first condition (1) is met for at least two continuous readings (≥ 20 min). In other words, the *dAH/dt* inside the *shaft* represented the AH changes in the *shaft* over time. This AH change was controlled by the air inflow or outflow from the upper or lower boundary, respectively. Therefore, *dAH/dt* was used to classify the airflow direction. The threshold value that was found to be the optimal to classify airflow was 50 % of the *dAH/dt* standard deviation. Using greater threshold values resulted in an underestimation of the number of both the inflow and outflow events, and vice-versa for the case of lower threshold values. As an example for the classification algorithm, results from the inflow and outflow classification for eight arbitrarily selected representative days (from Fig. 2) are shown in Fig. 3.

To identify the physical parameters that control the transition between the air inflow and outflow events, we focused and analyzed in details single events, one of them given as an example in Fig. 4. In a typical event, with both inflow and outflow, three stages were observed: (1) transition of *dP_atm/dt* (i.e., changes of barometric pressure over time) from negative to positive values (stage 1); (2) followed by a momentary decrease in temperature in the *shaft* observed by the temperature sensors at depths of 12 and 27 m (stage 2); and (3) finally inflow of air from the ambient atmosphere downward into the *shaft* that reduced the AH (stage 3). These stages were repeated in reverse in an outflow event (stages 4-6). The time lags between the changes of *dP_atm/dt* (stage 1) and the start of inflow/outflow events (stage 3) were similar in 60 % of the events (≤20 min). In the remainder 40 % of events, the time lags were greater than 20 min (20-60 min; Fig. 5).

The middle-stage of a momentary temperature change inside the *shaft* (stages 2 and 5 in Fig. 4) was previously reported by Perrier et al (2002). These investigators related the temperature changes to vertical movements of cold air plumes. In addition, these changes also provide further indication for the airflow origin. In an inflow event, it is expected that the sensor located closer to the atmosphere will respond before the lower one and the opposite in a case of an outflow from the cavity. Indeed, in an inflow event the negative *dT/dt* peak occurs at depth of 12 m before 27 m and vice-versa in an outflow event (Fig. 4a -3). Moreover, the fact that the temperature changes at each depth were correlative to the airflow direction as presented by the transition from stage 2 to 3, provides the ability to use simple, low-budget temperature sensors to estimate the direction of the airflow in the *shaft*.





Figure. 6 examines the correlation between the direction of air transport (inflow/outflow) and the atmospheric forcing, i.e., changes in barometric pressure and thermal stability. The general distribution of the $dP_{atm}/dt$ in the 42-days of measurements was defined as a normal distribution with an average $dP_{atm}/dt$ ~0 (Fig. 6a). However, when analyzing only the data set from the inflow/outflow events there was a clear trend in the $dP_{atm}/dt$ distribution. As expected, an inflow event (i.e., stage 3 of

AH decrease) occurred when barometric pressure increased, $dP_{atm}/dt > 0$ (Fig. 6b), whereas an outflow event (i.e., stage 6 of AH increase) occurred when barometric pressure decreased, $dP_{atm}/dt < 0$ (Fig. 6c). A similar analysis was done to check whether there is a correlation between the *shaft*-atmosphere temperature differences and the direction of air transport (i.e., AH changes). In this case, no significant changes were observed between the distribution in an inflow or outflow event (Fig. 6e and f) and the general distribution over the 42 days (Fig. 6d). The finding that $dP_{atm}/dt$ rather than temperature differences

explains a very large portion of the air transport polarity variance, implies that the inflow/outflow changes inside the *shaft* were due to atmospheric barometric pressure changes and not due to thermal instability (i.e., TIC) inside the *shaft*. Noteworthy is that because of the time lag (about 20 min) between the changes in $dP_{atm}/dt$ and the initiation of the inflow/outflow events, it is expected that not all inflow/outflow events will be explained by the $dP_{atm}/dt$ distribution analysis in Fig. 6. Previous studies where TIC was found to be the governing convective air movement mechanism in the subsurface

were focused on shallow systems (1-2 m depth; Ganot et al., 2014, 2012; Levintal et al., 2017; Weisbrod et al., 2009). Here, we focus on deeper settings where apparently barometric pressure variations are more important than TIC for the development of convective air movement.

Our results indicate that changes in atmosphere barometric pressure determine the advective airflow direction. This is presumably due to the difference between the barometric pressure and the cavity pressure caused by the vertical propagation

of the barometric pressure (Neeper, 2003; Neeper and Stauffer, 2012; Perina, 2014; You et al., 2011). In case of positive barometric pressure changes (i.e., increase of barometric pressure over time), the barometric pressure will be greater than the cavity pressure and thus inflow of air will develop. In contrast, outflow of air will start when negative pressure changes are present (i.e., decrease of barometric pressure over time).

### 3.2 *Borehole* geometry

Atmospheric conditions during the *borehole* measurements presented daily temperature and RH cycles (Fig. 2b -1 and 2, black lines). During daytime, air temperatures and RH values were 20-25 °C and 50-70 %, respectively, compared to night time values of 15-18 °C and 80-90 %. At a depth of 12 m, the average temperature, RH, and AH were 23.9 ± 0.3 ºC, 83.7 ± 13.0 %, and 18.2 ± 3.0 g/m$^3$, respectively (Fig. 2b -1, 2 and 3, purple lines). In contrast, at a depth of 27 m, measurements were significantly more stable than at 12 m: temperature, RH, and AH averages values were 25.1 ± 0.2 ºC, 90.0 ± 3.6 %, and

20.8 ± 0.9 g/m$^3$, respectively (Fig. 2b -1, 2 and 3, green lines).



In the inflow events the AH values in the middle *borehole* sensor (12 m) were similar to the upper atmospheric values (Fig. 2b -3, purple and black lines), whereas in the outflow events they equalled to those in the lower part of the *borehole* (Fig. 2b -3, purple and black dash lines). Nonetheless, the airflow effect was observed only at the 12 m depth and not at 27 m (Fig. 2b -3, green line). Therefore, we conclude that inflow events reached the depth of 12 m but did not reach depths of 27 m.

**3.3 Comparison between *shaft* and *borehole* geometries**

While all *shaft* and *borehole* parameters other than the connection to the lower cavity were identical and the atmospheric conditions in the two measurement periods were similar, there were clear differences between the two geometries. The *borehole* temperature readings at 12 m exhibited only half of the standard deviation compared to the same depth in the *shaft* ($\pm$ 0.3 $^o$C compared to $\pm$ 0.7 $^o$C). At 27 m, the differences between the two geometries were even more pronounced (Fig. 2a -

1 and 2; compare to b -1 and 2, green lines). No significant changes in temperature or RH along the measurement period were measured at 27 m for the *borehole*, while there were changes observed in the *shaft* (e.g., *shaft* RH at 27 m fluctuated between 60 and 100 %).

The *shaft*/*borehole* differences at the 27 m sensor can be explained using a simple two-reservoir model. In the case of the *shaft*, we can define both the atmosphere and the cavity as two- infinite air reservoirs connected via a finite volume *shaft*.

Therefore, air transport between the two reservoirs is unlimited and controlled only by the boundary conditions (i.e., barometric pressure). In this case, all sensors throughout the *shaft* will show similar AH decrease/increase, as evident from the similar changes of the purple and green lines in Fig. 2a -3. On the other hand, in the *borehole*, there is only one upper infinite reservoir (i.e., the atmosphere) and each inflow air transport is limited by the soil resistivity at the lower boundary (i.e., the soil capability to enable penetration of air inflow events – the soil permeability and porosity). Here, the effect of AH

decreases with depth and indeed the deepest sensor of 27 m showed no change in AH compared to the changes in AH at the 12 m depth (Fig. 2b -3).

**3.4 Comparison between *shaft*/*borehole* geometries and *large-diameter borehole***

Results from the *large-diameter borehole* are presented in Fig. 7. In contrast to the *shaft*/*borehole*, AH changes inside this *large-diameter borehole* (measured at 10 m depth) were correlated to the temperature differences between 10 m depth and

the atmosphere and not to the barometric pressure changes. No barometric pressure effect on the AH was observed, and during most of the measurement period the atmospheric AH and that at 10 m were similar (Fig. 7a -3, purple and black lines). This was because thermal-instability inside the *large-diameter borehole* (i.e., cold atmospheric air above warm borehole air) initiated TIC mixing with the atmosphere. The only cases of increase in AH at 10 m occurred when thermal-stability controlled the air movement inside the *large-diameter borehole* during noontime, and only then did AH at 10 m

increase to the saturation conditions that characterize the lower boundary (marked as gray columns in Fig. 7a). The



correlation between the positive temperature differences and the decrease in AH in this *large-diameter borehole* (3.4 m), which was not found in the small-diameter *shaft/borehole* (0.1 m), clearly indicated that TIC was the controlling mechanism for advective air movement in the *large-diameter borehole*. Furthermore, the fact that changes in barometric pressure did not lead to changes in AH, e.g., from 22/2 to 24/2 (Fig. 7a -4), suggest that BP was not the dominating driving force as in the

small-diameter *shaft/borehole*.

We posit that the main parameter controlling which transport mechanisms govern advective air movement is the *borehole* or *shaft* diameter. Given the same climate conditions, a small *borehole* diameter will decrease the magnitude of the TIC. This is because TIC magnitude in a cylinder geometry is positively proportional by the power of four to the cylinder radius (e.g., Berthold, 2010; Berthold and Börner, 2008; Berthold and Resagk, 2012; Rayleigh, 1916). Therefore, in our case of a small

diameter *borehole* of 0.1 m, TIC had minor influence on air transport inside the *shaft/borehole*.

In order to assess the *borehole* diameter ($r$) impact on the airflow, we compared the air velocity ($u$) dependency on the *borehole* diameter for the two gas transport mechanisms of BP and TIC. The analysis was done for a simple case in which the *borehole* perimeter is sealed except a perforated lower part, i.e., a cased *borehole*.

For BP, under the assumption of uni-dimensional cylindrical flow, the volume flow rate per unit length ($A$) is

approximately proportional to one-sixth power of the *borehole* radius (Eq. (2)), such that increasing $r$ by a factor of 10 will only increase $A$ by 41 % (Neeper, 2003).

$$A \propto r^{0.15} \tag{2}$$

where $r$ is the *borehole* diameter [m] and $A$ is the volume flow rate per unit length [m$^3$/m/s]. Because $Q$ is proportional to $A$, we can also define:

$$Q \propto r^{0.15} \tag{3}$$

where $Q$ is the *borehole* total volume rate to the atmosphere [m$^3$/s]. For a cylindrical flow $u$ is defined as:

$$u = \frac{Q}{\pi r^2} \tag{4}$$

where $u$ is the air velocity [m/s]. Thus, from Eqs. (3) and (4), we can conclude that the proportion between $u$ and $r$ is:

$$u_{BP} \propto \frac{r^{0.15}}{r^2} \; or \; u_{BP} \propto \frac{1}{r^{1.85}} \tag{5}$$

which means that $u$ will decrease with the increase of $r$.



For the case of TIC, given the same assumptions above, the thermal instability number ($Ra$), which is an indicator for $u$, is proportional to the temperature gradient ($dT/dz$) and to $r$ by the power of four (Berthold, 2010; Berthold and Resagk, 2012; Rayleigh, 1916):

$$Ra = \frac{\alpha \times g}{D_T \times v} \times \frac{dT}{dz} \times r^4 \qquad (6)$$

where $Ra$ is dimensionless [-], $D_T$ is the thermal diffusivity [m$^2$/s], $\alpha$ is the thermal expansion coefficient [1/K], $g$ is the gravitational acceleration [m/s$^2$], $r$ is the characteristic length dependent on the geometry, also defined as the radius of the *borehole* [m], and $v$ is the kinematic viscosity of the air [m$^2$/s]. Applying Rayleigh-Benard models to *borehole* geometry relates $Ra$ number to Reynolds number ($Re$, Eq. (7)) and to $u$ (Eq. (8)) (Grossmann and Lohse, 2000; Perrier et al., 2005):

$$Re = 3.5 \times Ra^{0.446} \qquad (7)$$

$$u = \frac{Re \times v}{h} \qquad (8)$$

where $h$ is the characteristic length [m] equal to $r$. Substituting Eqs. (6) and (7) in Eq. (8) results in:

$$u = \frac{(0.35 \times (\frac{\alpha \times g}{D_T \times v} \times \frac{dT}{dz} \times r^4)^{0.446}) \times v}{r} \qquad (9)$$

because we are comparing BP and TIC under the same atmospheric conditions and *borehole* diameter, we can consider the following parameters as constants: $\alpha$, $g$, $D_T$, $v$, and $dT/dz$. Therefore, $u$ is proportional to $r$ such that:

$$u_{TIC} \propto r^{0.784} \qquad (10)$$

Finally, when comparing the $u$ dependency on $r$ for the case of BP (Eq. (5)) and TIC (Eq. (10)) it is clear that an increase in $r$ will have a contrasting effect on the airflow generated from TIC compared to the one generated from BP; increase in $r$ will increase the $u$ generated from TIC while decreasing the $u$ generated from BP. This quantitatively supports the conclusion from the field observations that in a *large-diameter borehole* TIC was more significant to the gas transport than BP.

The use of Eqs. (5) and (10) for comparison purpose cannot be addressed without considering the differences of $u$ between BP and TIC due to the flow geometry. In BP $u$ is uni-directional (inward or outward flow), whereas in TIC $u$ represents a bi-directional flow (e.g., Eckert and Diaguila, 1955). Nevertheless, in both cases (BP and TIC) $u$ describes the same physical meaning of air velocity magnitude. Thus, we still consider Eqs. (5) and (10) as a good first order approximation for comparing the correlation between airflow and *borehole* radius for BP and TIC conditions.

It should be emphasized that the threshold value of $r$ to determine when TIC dominates BP and vice versa cannot be considered as one absolute value. This is because atmospheric conditions differ between different sites, thus the magnitude





of $dT/dz$ and $dP_{atm}/dt$ can change. For example, a tropical climate will exhibit a smaller diurnal temperature cycle, which will cause a lower $dT/dz$. Therefore, TIC intensity will be markedly suppressed compared to the same *borehole* in an arid climate.

A conceptual model was developed to present the advective transport mechanisms of the three geometries (Fig. 8). The differences between the *borehole* and the *shaft* are illustrated in Fig. 8a and the differences between them and the *large-diameter borehole* in Fig. 8b. The *borehole* diameter will define which advective transport mechanism is more significant: at a small diameter of 0.1 m, BP controls gas transport (Fig. 8a), whereas at the *large-diameter borehole* of 3.4 m TIC is the dominant mechanism (Fig. 8b).

**3.5 Field implications**

The mechanisms controlling the subsurface-atmosphere air exchange have several important implications. These include, for example, volatile organic compounds (VOCs) transport from the subsurface to the atmosphere in contaminated sites (Boothroyd et al., 2016), natural aeration (oxygen supply) of underground quarries or tunnels and the need for artificial, enhanced, air exchange facilities in such environments and changes of RH values in karst systems. For example, RH changes in a mine underground atmosphere have great influence on the rock physico-mechanical parameters and stability (Auvray et al., 2008). Commonly used mine *shaft*s can induce rapid RH changes at the *shaft*-cavity interface as presented above, which can then lead to rock stability problems. Shafts can also be used for fast removal of water vapour from deep soil layers, in order to lower its hydraulic conductivity and subsequently cease downward transport of contaminants.

One of the important implications is the potential role of *shafts* and *boreholes* as conduits for air-exchange to the overall GHG emission and related mechanisms such as carbon capture and storage processes (CCS). Because the *borehole* geometry ($r \sim 0.1$ m) is more common than the *large-diameter borehole* geometry ($r \sim 3$ m), we will focus on the *borehole* for the overall GHG transport discussion. Two basic assumptions are here to consider: first that the BP air transport rate is up to a few-order of magnitude greater than diffusion, (You et al., 2011), and second that these conduits can act as "pipes" to the Earth's subsurface, connecting elevated GHG sources, such as $CO_2$ or Methane, to the atmosphere.

To make a first order evaluation of the potential $CO_2$ emission from our *borehole*, we measured the $CO_2$ within the two sites for two weeks during the winter season 2017. Results showed that during outflow events $CO_2$ concentrations throughout the *borehole* were up to ~2000 ppm (data are not shown). Without getting into specific calculation of $CO_2$ mass flux that depends on numerous parameters and likely to vary in different environments, this validates that the *borehole* geometry acts as a source for atmospheric $CO_2$ rather than a sink.

A more significant GHG type emitted from *borehole* is Methane, which its emissions were quantified for 19 *boreholes* in Pennsylvania (Kang et al., 2015, 2014). After upscaling their results to the state level, it was proposed that these *boreholes* emissions represent 4–7% of the total methane emissions in Pennsylvania. Their research focused mainly on the production





function of Methane and not on the physical transport mechanism. Implementing our conclusion that BP was the main air transport mechanism can indicate that the Methane emissions presented by Kang et al. (2015, 2014) occurred mainly during periods of $dP_{atm}/dt < 0$.

## 4 Conclusions

Three borehole geometries were compared to explore air transport mechanisms under natural, variable, atmospheric conditions. The first case was a 27 m vertical *shaft* with a 0.1 m diameter that connected a large underground cavity to the atmosphere, the second case was the same *borehole* but connection to the underground cavity was blocked and the pipe ended in the unsaturated soil matrix. The third was a *large-diameter borehole* of 3.4 m in diameter and 59 m depth. In the first two, *shaft* and *borehole*, the air inflow and outflow at 12 m were found to be correlated to the changes in barometric

pressure (BP). However, in the *large-diameter borehole*, the air transport at a similar depth (10 m) was correlated to thermal-instability (TIC) rather than barometric pressure.

Use of AH changes was shown as a practical tool to identify the source of air parcels within the three geometries, namely atmospheric vs. lower-borehole/cavity, and thus to determine the direction and effect of the air transport. Water vapor concentrations in the atmosphere vary along the day, while are almost constant in underground cavities, and therefore can be

used as a natural tracer for air source and flow directions without injecting additional gases.

A conceptual model is presented to describe the induced airflow in all three geometries. In the *shaft*, the atmospheric air entered through the *shaft* to the cavity and vice-versa. In other words, the *shaft* connects between two large air sources and inflow and outflow via the *shaft* is determined according to the barometric pressure changes. In the *borehole*, the atmospheric air entrance was limited by the soil resistivity at the lower boundary. Thus, inflow of atmospheric air was

observed only at 12 m depth and not at the deeper 27 m sensor. BP was found to control air advective transport in both geometries. On the other hand, in the third geometry of a *large-diameter borehole*, thermal-instability initiated TIC advection while BP did not play a significant role. This caused circulation of atmospheric air into the *borehole* to a depth of 10 m, whenever the thermal instability occurred.

In summary, our observations improve our understanding of the governing mechanisms controlling air movement in

boreholes and shafts as a function of their geometries and diameters as well as the ambient atmospheric conditions. Understanding these mechanisms is important for shedding light onto air dynamics in cavities (e.g., caves, underground storage structures, quarries, abundant boreholes, observation pipes, tunnels, etc.). In addition, our observations assist to better calculate GHG fluxes from these domains as well as estimate the time periods when these fluxes are enhanced.





**Data Availability.** The data set used in the analyses is public, and available from https://doi.org/10.6084/m9.figshare.5786796.v1 and in supporting information.

**Author Contributions.** EL, NGL, AM and NW performed the data analysis. EL and NW wrote the first draft of the manuscript and all authors contributed to the final version.

5 **Competing Financial Interests.** There are no competing financial interests.

**Acknowledgements.** This work was funded by the Israeli Science Foundation (ISF), contract 678/11, The Bi-National Science Foundation (BSF) contract number (2014220), and the Israeli Ministry of Agriculture, contract 857-0686-13. We also acknowledge the Sam Zuckerberg scholarship provided to EL. The field observations were conducted with the Geological Survey of Israel team: Hallel Lutzky, Uri Malik, Haim Chemo, Ziv Mor and Haggai Eyal; and Raz Amir from 10 the Ben-Gurion University of the Negev.

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



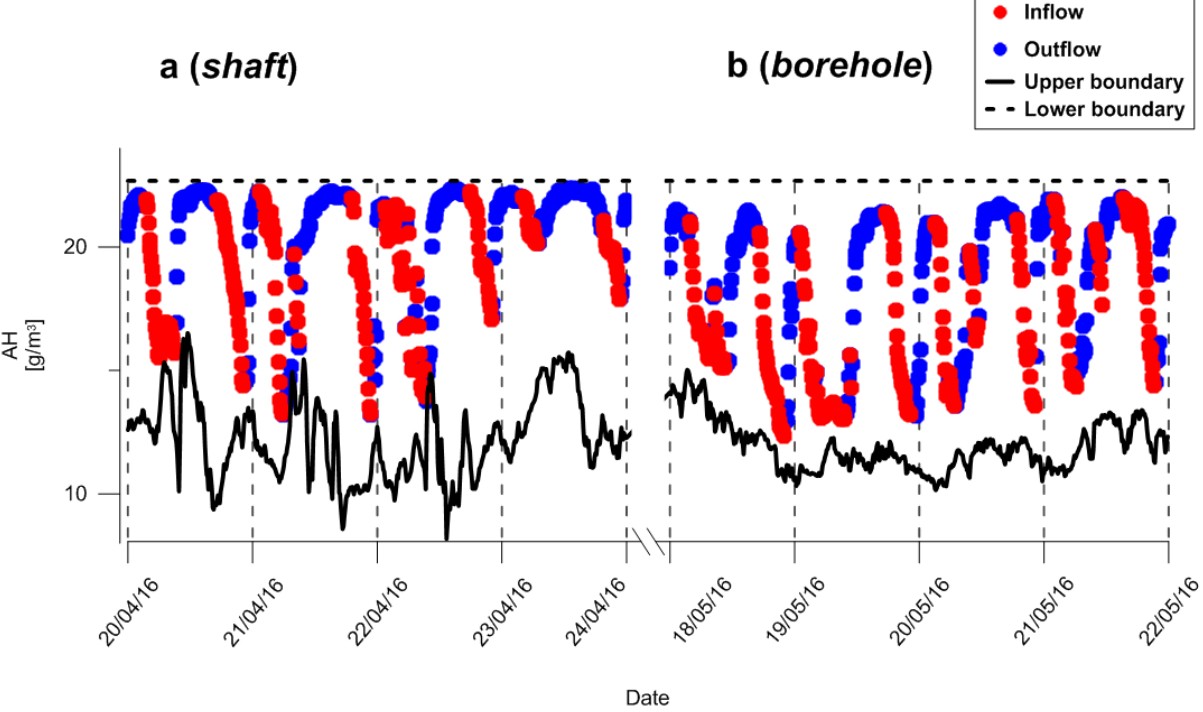

**Figure. 3. Classification of inflow and outflow transition events at 12 m depth for *shaft* (a) and *borehole* (b). Upper boundary represents the AH values according to the temperature and RH measured at 2 m above ground; lower boundary represents the AH values in the underground cavity (*shaft*) or the soil-*borehole* interface (*borehole*). Point intervals are at 10 min each.**



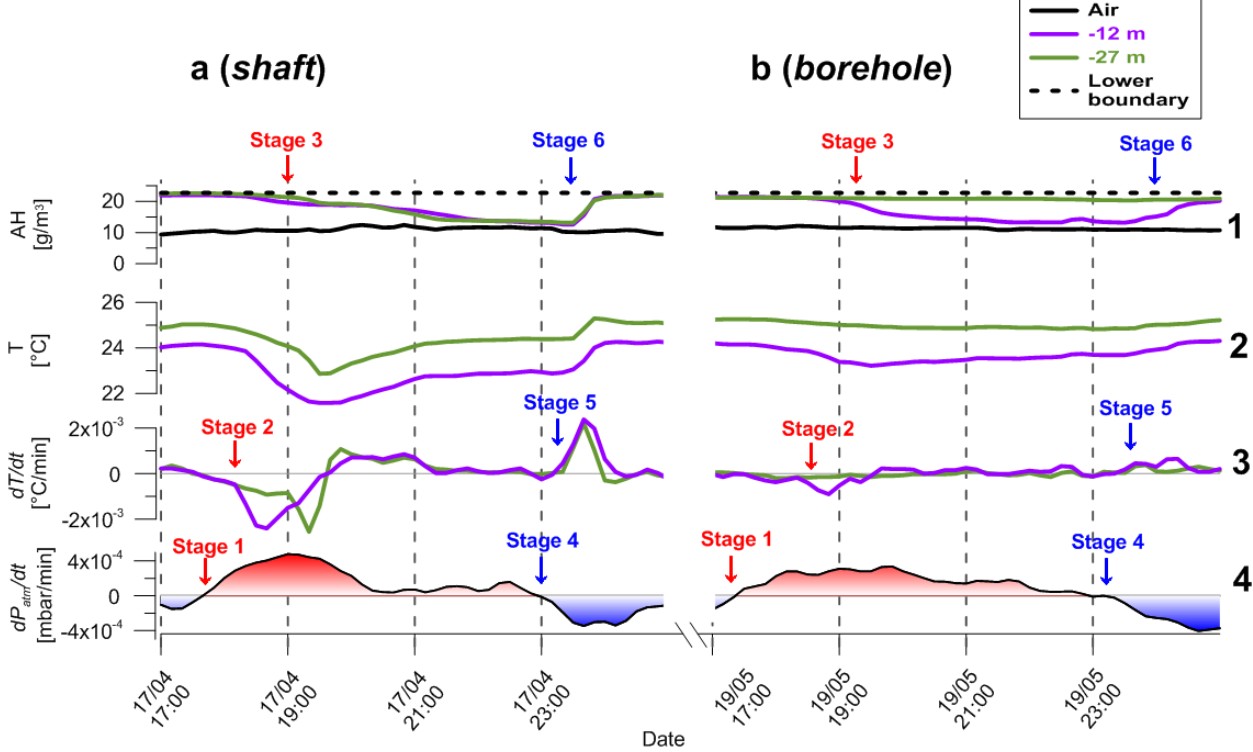

**Figure. 4. Time series results from a single transition event. Left side (a) and right side (b) represent the *shaft* and *borehole* results, respectively. Red and blue text fonts represent stages in the inflow and outflow events, respectively. *dT/dt* values were derived from the temperature values in line 2.**





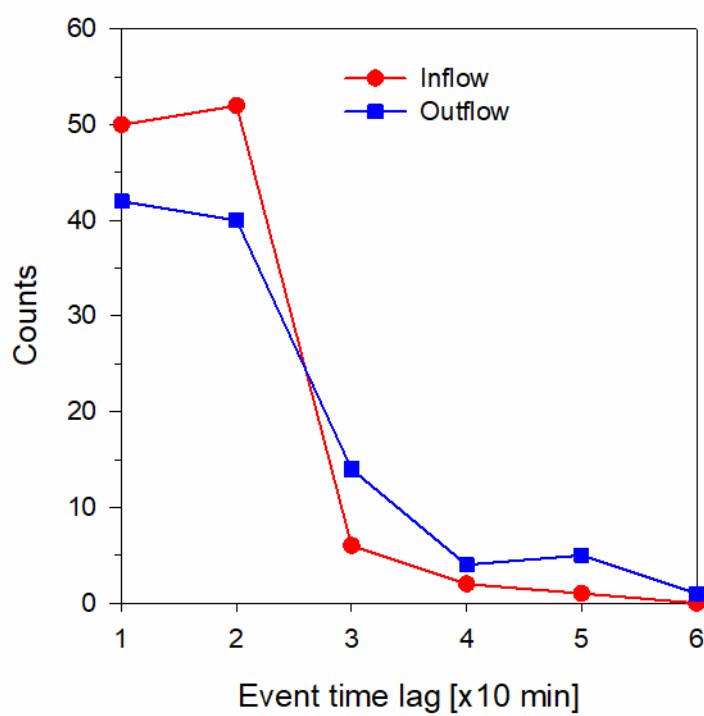

**Figure. 5. Time lags between changes of $dP_{atm}/dt$ (i.e., changes of barometric pressure over time) and the start of inflow/outflow events. Event classification was done automatically using the data from the 12 m depth sensors. Red symbols represent the time lag between the transition of $dP_{atm}/dt$ from negative to positive values and the start of an inflow event (stages 1 and 3 in Fig. 4). Blue**

5      **symbols represent the time lag between the transition of $dP_{atm}/dt$ from positive to negative values and the start of an outflow event (stages 4 and 6 in Fig. 4).**



**Figure. 6. Histograms of changes of atmospheric barometric pressure ($dP_{atm}/dt$) and temperature difference between the *shaft* and the atmosphere. The gray color (a, d) represents data from all 42 days of measurement. Red (b, e) and blue colors (c, f) represent data from the inflow and outflow events, respectively. Positive values of $dP_{atm}/dt$ can drive inflow events from the atmosphere into the underground cavity, whereas negative values can drive outflow events. Temperature differences (X-axis) are between 12 m-**

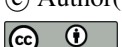



deep inside the *shaft* and the atmosphere, meaning that positive values represent cases in which the *shaft* was warmer than the atmosphere (i.e., thermal instability) and vice versa.

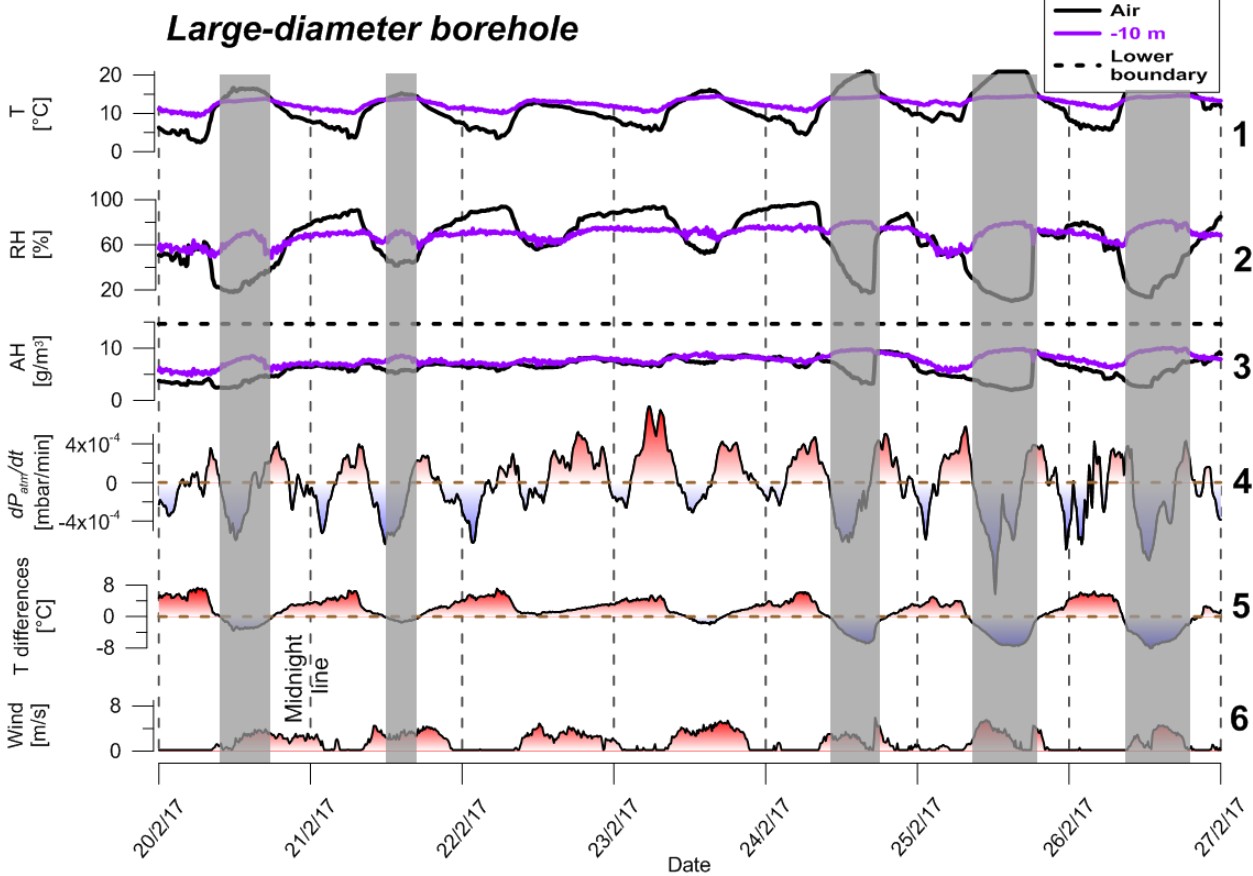

Figure. 7. Time series results from the *large-diameter borehole* for one week. Gray columns represent periods of thermal-stability inside the *large-diameter borehole*. Values in line 5 represent the temperature differences between the sensor at 10 m depth and the sensor above ground.




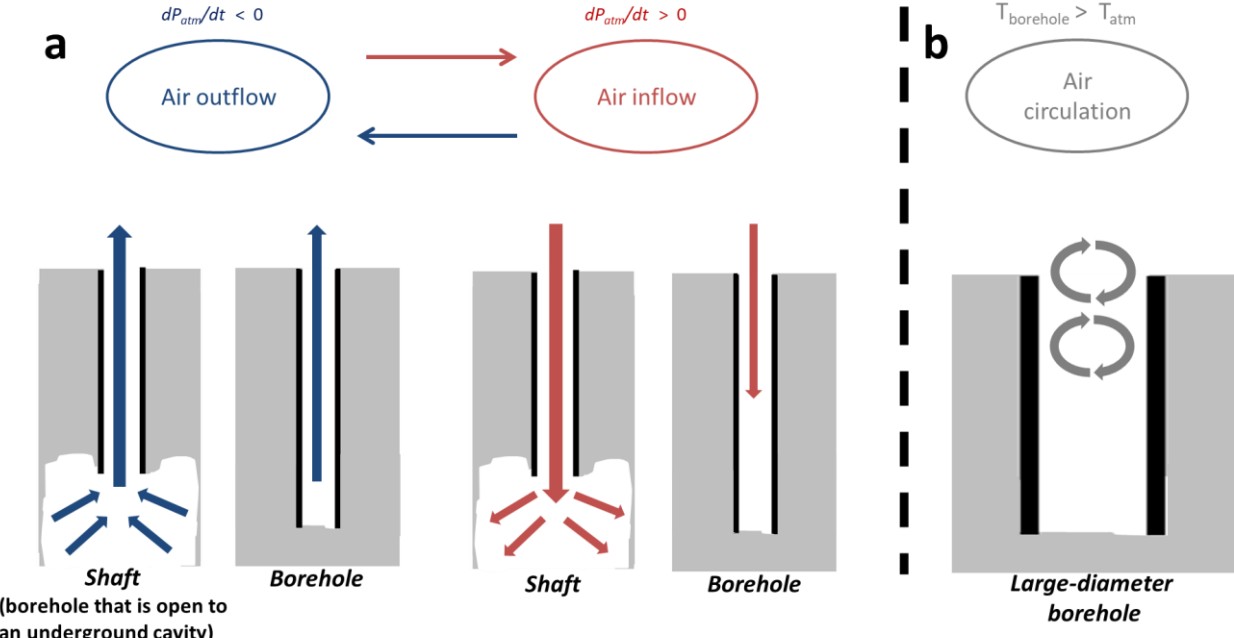

**Figure. 8. Conceptual model for airflow inside boreholes and shafts. Left side (a) represents the *shaft/borehole* and right side (b) represents the *large-diameter borehole*. Blue arrows illustrate the air outflow events in which air flows from the bottom cavity (*shaft*) or the bottom *borehole*-soil interface (*borehole*) to the atmosphere, whereas red arrows illustrate the air inflow events in the opposite direction. For example, as the red arrows indicate, air from the atmosphere will enter the cavity, equalling the absolute humidity (AH) values throughout the *shaft* to the atmospheric values. In contrast, in the *borehole*, this stage will only be effective to a certain depth and the bottom *borehole*-soil interface will not be significantly affected. In the *large-diameter borehole*, gray arrows illustrate circulation of air from the atmosphere into the *large-diameter borehole* due to thermal-instability that initiates TIC. The diameter will define which advective transport mechanism is more significant: at a small diameter of 0.1 m, BP controls gas transport (a), whereas at a larger diameter of 3.4 m TIC is the dominant mechanism (b).**