# Peer review of "Supporting information for"

_Earth System Dynamics, 2018_

## Referee Comment (RC1) · A. Kowalski (Referee) · 4 Jun 2018

Levintal and colleagues have measured the state and composition of air in two boreholes (one connected at times to a large, underground cavity) over some brief periods and tried to generalise about transport processes through these "pipes". Such a study could be a welcome addition to the literature regarding the dynamics of vadoze-zone air and its exchange with the atmosphere, about which little is still known. However, the paper suffers from some significant shortcomings, both technical and in terms of perspective, that should be improved before the paper can be acceptable for publication. Therefore, I recommend major revision along the following lines.

[Figure]

General comments

Technical comment on air density

A major theme is thermal-induced convection (TIC), which the authors attribute to "unstable density gradients resulting from temperature differences". This is not quite correct. At a given height/pressure, air density depends predominantly on temperature but also on air composition. This can be important particularly when dealing with such a humid cavity. As a simple example with data taken from inspecting Fig. 2, atmospheric air at 25°C with 50% relative humidity and 400 ppm of $CO_2$ is warmer but denser than saturated cavity air at 24°C with 2000 ppm of $CO_2$. The authors cite Sánchez-Cañete et al. (2013) but apparently without having appreciated the message of that paper. Rather than the temperature, it is the virtual temperature (easily calculated using an excel spreadsheet offered by Sánchez-Cañete and colleagues, and that can be downloaded from http://fisicaaplicada.ugr.es/pages/tv/!/download) whose gradient should be examined to diagnose convective instability. Note that the virtual temperature depends predominantly on the temperature, secondarily on humidity, and tertiarily on the $CO_2$ concentration. The $CO_2$ dependence is only of relevance for the high $CO_2$ concentrations found in some underground environments.

Perspective on $CO_2$ concentrations

The abstract cites "High $CO_2$ concentrations (∼2000ppm)". In underground environments, $CO_2$ concentrations of over 100,000 ppm have been measured (Amundson and Davidson, 1990, J Geochem Explor, 38, 13-41), and values exceeding 10,000 ppm are not uncommon (Denis et al., 2005, Geophys Res Lett, 32, L05810, doi: 10.1029/2004GL022226; Benavente et al., 2010, Vadose Zone J. 9:126–136; Sánchez-Cañete et al., 2010, Geophys Res Lett, 38, L09802, doi:10.1029/2011GL047077). In this context, 2000 ppm is not high, and might even qualify as particularly low. Unlike that found in many caves, it is even too low to affect air density (but note that the water vapour content of the cavity certainly does, which is

why the virtual temperature is needed here).

Perspective on seasonal variability of atmospheric conditions

The manuscript presents borehole/shaft observations under atmospheric conditions that (1) are limited to a few weeks for each geometry and (2) vary seasonally among geometries, but the analysis fails to take into account these data limitations when generalizing about findings.

For example, the abstract claims that "absolute humidity was found to be a reliable proxy for distinguishing between atmospheric and cavity air masses and thus to explore air transport through the three geometries". This may be so for the spring (borehole, shaft) and mid-winter (large-D borehole) data in the paper, when atmospheric absolute humidity is considerably below that of the underground environment. However, in summer it is likely that atmospheric values of absolute humidity could equal, surpass, or oscillate about the subterranean value, and this proxy variable could well lose all of its utility. The same is true about the ability to determine shaft airflow directions using temperature sensors (p5, line 28).

In Section 3.4, the authors contrast *winter* air movement detected in the large-D borehole with those noted in the shaft/borehole during *springtime*, when the external air has likely become far less dense. Convective instability for these "pipes" must generally be far greater for either geometry in February than in May. Nonetheless, the ensuing analysis suggests that dT/dz can be considered constant. There is no justification for this spurious assumption, particularly given that the authors have data that enable its direct assessment. I have little doubt regarding the authors' conclusion that convection is most important for the large-D borehole, whereas barometric changes dominate for the other geometries, but this needs to be demonstrated climatologically and without making invalid assumptions.

The paper would be greatly strengthened by broadening the dataset to include a seasonal range of temperatures/humidities for every geometry. Otherwise, the generalisations inferred by the authors should be reduced in scope to take into account their seasonal representativeness.

Specific Comments

In addition to the General comments above, which require significant revision to the analysis and presentation of the findings, I offer the following specific suggestions (by page/line) to improve the manuscript:

2/16 Do not neglect differences in air composition.

3/11 Homogenize the font size "first 42 days"

3/13 That "the pipe touched the cavity floor" does not ensure a hermetic seal. Either the cavity floor geometry should be described (horizontal, smooth?), or the isolation of the pipe from the cavity should be somehow justified.

4/1 In equations, it is preferable that each variable be represented by a single symbol, with necessary subscripts. A reader might interpret "AH" in equations (1) and also in the derivative at page 5, line 5 (and elsewhere) as the product of the variables "A" (volume flow rate per unit length) and H. Rather than "AH", I recommend the use of the standard rho (greek symbol for "r") with a subscript (v) to indicate that it is the vapour density that is synonymous with the absolute humidity. Similarly, relative humidity could be represented with "U" instead of "RH".

5/16-19 The "typical" situation described is valid only for nighttime data (and is furthermore dependent upon season). Thus, a "decrease in temperature in the shaft observed by the temperature suensors" would not typically describe penetration during a daytime barometric increase since the external air temperature would exceed that of the cavity.

5/28-30 The ability to estimate the direction of the airflow in the shaft is not general. For example, a pressure change that occurred at a moment when the external air temperature coincided with the cavity temperature (as occurs twice per day in Fig. 2, and perhaps during many hours on summer nights) would not be detected.

6/24 Homogenize header format.

7/29 These cases that occur only at noontime are specific to winter, when the air temperature rarely exceeds the cavity temperature. In summer, it would be most often the case.

8/6-7 The validity of this hypothesis is very dependent on season. "Given the same climate conditions..." I agree, but the paper does not examine these geometries under the same climate conditions.

8/11-etc Much of the following 1.5 pages constitutes methodology, and would better be provided in section 2.

9/13 The assumption that dT/dz is constant makes no sense. Although BP and TIC are compared using the same climate conditions, TIC depends on dT/dz whereas BP does not. Increased values of dT/dz in winter make for greater convective instability, whatever the borehold diameter. But the geometries are not examined under the same climate conditions, since the shaft and borehole are examined in spring, whereas the large-D borehole is examined in winter.

10/3 The conceptual model is a very good idea.

10/23-27 This paragraph regarding CO2 source/sink behaviour is particularly weak, and I recommend deleting it entirely. To speak of sources/sink requires information regarding the origin/destiny of the CO2 in the cavity. In any event, I think it is absurd to state that the geometry is a source or sink. It might better be described as a storage space for CO2 that could otherwise have been emitted to the atmosphere (but from what source?). To appreciate the complexities of the source/sink issue, the authors are encouraged to consult section 3.1 of Serrano-Ortiz et al., 2010, Agric. and Forest Meteorol. 150, 321–329.

11/12 This "conclusion" is very much specific to site and season, and must be qualified as such or revised.

---

## Referee Comment (RC2) · Anonymous Referee #2 · 25 Jun 2018

Earth's atmosphere is extremely complex system oneself and its interaction with near-surface targets, deep dynamic geological-geophysical regularities, and some cosmic factors (e.g., tidal effects) increases the total complexness.

Without hesitation, Levintal et al. have arisen very important problem of interaction between the underground caves, boreholes and mines with the Earth's atmosphere. This publication obviously will trigger a series of new publications in this field.

For instance, in the world a lot (tens of millions) of comparatively deep (> 500 m) boreholes were drilled in different physical-geological environments. Many of them are open, semi-open or have indirect connection with the Earth's atmosphere. Calculation

of the total effect of air transport from these objects is a difficult physical-mathematical problem.

Some minor remarks are compiled below.

I believe that 'boreholes' and 'mines' cannot be included to the class of 'caves' since they are principally different targets. Besides this, most part of caves are the natural geological objects existing sufficiently long time, whereas boreholes and mines are the artificial targets which have been appeared mainly in 20th century.

Generally speaking, examination of two (three ?) targets only is insufficient one. I propose that general conclusions done in this MS for all types of underground objects is untimely one.

$CO_2$ concentrations in various underground targets strongly exceed the value of 2000 ppm (e.g., Guillon et al., 2015).

I can suggest that the role of viscosity in air transport (Finkelstein et al., 2006) may be more significant than presented in the MS.

The authors assumed some physical parameters as constant (for simplicity of calculations). It is a widely distributed approach and it is acceptable, for instance, for gravity acceleration and thermal expansion. However, accepting viscosity as constant is under question (Finkelstein et al., 2006). From numerous thermal measurements in wells follows that the behavior of dT/dz is not constant one (e.g., Huang et al., 2000; Eppelbaum et al., 2014). It should be taken into account in the further extension of this approach.

Obviously, an interaction between the near-surface targets and Earth's atmosphere has nonlinear character (e.g., Kardashov et al., 2000). It cannot be realized in the presented study, but can be reflected in future investigations.

I propose that after a small revision, this MS may be accepted for publication.

**References**

Eppelbaum, L.V., Kutasov, I.M. and Pilchin, A.N., 2014. *Applied Geothermics*. Springer, Heidelberg – N.Y.

Finkelstein, M., Eppelbaum, L. and Price, C., 2006. Analysis of temperature influences on the amplitude-frequency of Rn gas concentration. *Journal of Environmental Radioactivity*, **86**, No. 2, 251-270.

Guillon, S., Agrinier, P. and Pili, E., 2015. Monitoring $CO_2$ concentration and $\delta^{13}C$ in an underground cavity using a commercial isotope ratio infrared spectrometer. *Applied Physics B*, **119**, No.1, 165-175.

Huang, S., Pollack, H.N. and Shen, P.Y., 2000. Temperature trends over past five centuries reconstructed from borehole temperatures. *Nature*, **403**, Feb. 17, 756-758.

Kardashov, V.R., Eppelbaum, L.V. and Vasilyev, O.V., 2000. The role of nonlinear source terms in geophysics. *Geophysical Research Letters*, **27**, No. 14, 2069-2073.

---

## Referee Comment (RC3) · Anonymous Referee #3 · 5 Jul 2018

This paper, after a noteworthy revision, could be a useful contribution to the literature on air transport and borehole-atmosphere exchanges. This manuscript studies the air transport between three borehole types using temperature and pressure gradients and using the water vapor as tracer of air flow. Levintal et al. found that the main mechanism driven the air transport depend on the geometries, where changes in pressure induce more transport in narrow boreholes and temperature differences induce more transport in wide-diameter boreholes.

My mayor comment to this paper is that the authors are studying the air transport neglecting the air composition to estimate the air mass buoyancy. The virtual temperature

(Tv) is the temperature at which dry air would have the same density as the moist air, at a given pressure. In other words, two air samples with the same Tv have the same density, regardless of their actual temperature or relative humidity. Levintal et al are measuring the relative humidity in the external and internal air, therefore they would be estimate the virtual temperature to study the buoyancy.

The authors mention that they found around 2000 ppm of $CO_2$, however they don't show the $CO_2$ pattern in any graph and the $CO_2$ sensors are not described in methodology. I suggest showing the data as supporting information and discuss the possible influence in the air composition and its buoyancy.

Minor comments (line/page):

1-5/3: How far are both sites? Could you include coordinates?

11/3: Edit "42".

30/3: It's not clear how many sensors do you have and their positions? Please add more information and include this information in Fig 1b.

8/4: delete "An example of"

13/4: the number 6050 will be 6048 (6 measurements/h*24h/day*42day)

14-15/4: provide the % of relative humidity for 12 and 27 separately. In figure 1 and figure S1 I can see that during the whole period the relative humidity is always higher at 12m than at 27m, does that make sense?

9/7: Could it be because the max-min variation in temperature is higher in the shaft than in the borehole?

24/7: could you provide some analysis (as a simple $R^2$) to prove that the correlation AH-Temperature is higher than AH-pressure?

3-5/8: I can see in Figure 7 that negative values on dPatm/dt increase the RH at 10m.

During the days 22-24 probably the atmospheric pressure was changing from low to high pressure and for this reason the negative values on dPatm/dt were much lower. Would be useful if you show the atmospheric pressure value in a second Y-axis in Fig7, panel 4.

5-end/11: In your conclusions, you would remark that these conclusions were carried out only with data during 42 days in the shaft, 4 days in the borehole and 7 days in the large-diameter borehole, therefore, in the future we need investigate during longer periods, ...because for example in the case of the $CO_2$, you found 2000 ppm in spring, but commonly the maximum values of $CO_2$ are reached in summer/fall and therefore they could affect to the buoyancy.

Figure2: In the legend "cavity air" is the Lower boundary, isn't it?

2-3/17: delete "representative"

4/17: change from "black dashed line in 3" to "...in panel 3"

Figure 7: increase the scale on panel 1 because the air temperature during the day 25 is chopped, and also move the legend box.

---

## Author Comment (AC1) · 15 Jul 2018

**Pipes to Earth's subsurface: The role of atmospheric conditions in controlling air transport through boreholes and shafts**

Letter of response:

Reviewer 1:

Levintal and colleagues have measured the state and composition of air in two boreholes (one connected at times to a large, underground cavity) over some brief periods and tried to generalize about transport processes through these "pipes". Such a study could be a welcome addition to the literature regarding the dynamics of vadoze-zone air and its exchange with the atmosphere, about which little is still known. However, the paper suffers from some significant shortcomings, both technical and in terms of perspective, that should be improved before the paper can be acceptable for publication. Therefore, I recommend major revision along the following lines.

We would like to thank Prof. Kowalski for his comments and fruitful review. The manuscript was significantly changed. The main changes were: expansion of the dataset of the large-diameter borehole for the spring season; addition of a sensitivity analysis for the use of absolute humidity as a tracer for airflow direction according to half a year of measurements; and changes to Figs. 2, 6, and 7 according to the virtual temperature analysis that was suggested in this review. A detailed response for each comment is presented below in blue font.

**General comments**

Technical comment on air density

A major theme is thermal-induced convection (TIC), which the authors attribute to "unstable density gradients resulting from temperature differences". This is not quite correct. At a given height/pressure, air density depends predominantly on temperature but also on air composition. This can be important particularly when dealing with such a humid cavity. As a simple example with data taken from inspecting Fig. 2,

atmospheric air at 25_C with 50% relative humidity and 400 ppm of CO2 is warmer but denser than saturated cavity air at 24_C with 2000 ppm of CO2. The authors cite Sánchez-Cañete et al. (2013) but apparently without having appreciated the message of that paper. Rather than the temperature, it is the virtual temperature (easily calculated using an excel spreadsheet offered by Sánchez-Cañete and colleagues, and that can be downloaded from http://fisicaaplicada.ugr.es/pages/tv/!/download) whose gradient should be examined to diagnose convective instability. Note that the virtual temperature depends predominantly on the temperature, secondarily on humidity, and tertiarily on the CO2 concentration. The CO2 dependence is only of relevance for the high CO2 concentrations found in some underground environments.

We fully agree with this comment and after re-reading the manuscript by Sánchez-Cañete et al. (2013), we revised our MS accordingly:

(1)    A summary of the virtual temperature concept ($Tv$) was added to the introduction: "Although air density depends mainly on temperature, it is also depend on the air humidity, and to a lesser degree on the air's gas composition (Kowalski and Sánchez-Cañete, 2010). Integration of these three effects (temperature, relative humidity, and air composition) into a single parameter named virtual temperature ($Tv$) was proposed by Sánchez-Cañete et al. (2013)…. For a given altitude level, the $Tv$ differences will determine the onset of TIC." (Page 2, lines 20-27).
(2)    In Figures 2, 6, and 7, the data are now presented as a function of $Tv$ rather than T, according to the excel spreadsheet offered by Sánchez-Cañete et al. (2013).

We note that due to the fact that we didn't have continuous $CO_2$ measurements in this study, the $Tv$ within the boreholes was calculated using the T and RH continuous measurements with a constant $CO_2$ concentration according to preliminary results that we took over two weeks during the 2017 winter season  (~2000 ppm). Obviously, the $CO_2$ concentration varies throughout the day within the boreholes; however, we think that this is a reasonable choice because we know from these two weeks of measurements that the $CO_2$ inside the borehole did not increase above 2000 ppm, and the values are relatively low compared to other underground cavities such as caves (as also mentioned here in the reviewer's next comment). In addition, as emphasized by the reviewer in this comment,

the $CO_2$ changes are only the third parameter in importance after temperature and RH for air density convection, and therefore we believe this to be a reasonable assumption.

**Perspective on CO2 concentrations**

The abstract cites "High CO2 concentrations (2000ppm)". In underground environments, CO2 concentrations of over 100,000 ppm have been measured (Amundson and Davidson, 1990, J Geochem Explor, 38, 13-41), and values exceeding 10,000 ppm are not uncommon (Denis et al., 2005, Geophys Res Lett, 32, L05810, doi: 10.1029/2004GL022226; Benavente et al., 2010, Vadose Zone J. 9:126–136; Sánchez-Cañete et al., 2010, Geophys Res Lett, 38, L09802, doi:10.1029/2011GL047077). In this context, 2000 ppm is not high, and might even qualify as particularly low. Unlike that found in many caves, it is even too low to affect air density (but note that the water vapour content of the cavity certainly does, which is why the virtual temperature is needed here).

The word "high" was deleted from the last line in the abstract. We also added three of the above references with a suitable clarification that "In environments of high $CO_2$ concentrations compared to the atmosphere, the importance of the gas composition on the *Tv* becomes more pronounced. Such underground environments can be karstic areas of carbonate rocks (Sanchez-Cañete et al., 2011), caves (Denis et al., 2005; Guillon et al., 2015), and soils (Amundson and Davidson, 1990) where $CO_2$ concentrations can be very high, ranging from 10,000 to 100,000 ppm and above." (Page 2, lines 23-27).

**Perspective on seasonal variability of atmospheric conditions**

The manuscript presents borehole/shaft observations under atmospheric conditions that (1) are limited to a few weeks for each geometry and (2) vary seasonally among geometries, but the analysis fails to take into account these data limitations when generalizing about findings.

As we did measurements in the large-diameter boreholes for a longer time, we could change its time-series from February 2017 to April 2017 such that the analysis of both boreholes would be in the same season (i.e., spring). We note that there are still some

differences in the atmospheric conditions between the boreholes, this is because (1) the small diameter borehole was studied in 2016, whereas the large-diameter borehole was studied in 2017, and (2) the two boreholes are not exactly at the same geographical location. For example, the average atmospheric temperature and RH during April 2016 were 20.2° C and 69.1%, respectively, (at the small diameter borehole location) compared to the average values during April 2017 of 18.2° C and 50.4%, respectively, (at the large-diameter borehole location).

Unfortunately we don't have complete one-year results of the boreholes due to technical restrictions, and we agree that this is a limitation of our study. We added this limitation in the conclusions section: "…This mechanistic explanation was validated using the winter and spring season's dataset. Although we show that theoretically the transport mechanism observed for winter and spring should hold, with reduced significance for summer and autumn, further data are needed to verify the theoretical calculation." (Page 11, lines 26-28).

> For example, the abstract claims that "absolute humidity was found to be a reliable proxy for distinguishing between atmospheric and cavity air masses and thus to explore air transport through the three geometries". This may be so for the spring (borehole, shaft) and mid-winter (large-D borehole) data in the paper, when atmospheric absolute humidity is considerably below that of the underground environment. However, in summer it is likely that atmospheric values of absolute humidity could equal, surpass, or oscillate about the subterranean value, and this proxy variable could well lose all of its utility. The same is true about the ability to determine shaft airflow directions using temperature sensors (p5, line 28).

Following this comment, we broadened the absolute humidity dataset according to our available measurements and performed a sensitivity analysis of the changes in absolute humidity for the *large-diameter borehole* between 02/2017 and 08/2017 (see Fig. S2 and S3 in the Supporting information – also attached here below). A summary of the sensitivity analysis was added to the article to provide the limitation of using absolute humidity as a proxy for airflow in these types of boreholes:

(1) In the abstract we added: "Absolute humidity was found to be a reliable proxy for distinguishing between atmospheric and cavity air masses (mainly during winter and spring seasons), and thus to explore air transport through the three geometries." (Page 1, lines 14-15).

(2) In the discussion we added: "The use of AH as a proxy for airflow direction is suitable mainly for winter and spring seasons when the atmospheric AH is lower compared to AH within underground cavities (Figs. S2 and S3– supporting information). During the summer season, there are periods in which atmosphere and cavity AH are in equilibrium, and thus the use of AH as a proxy for airflow directions would not be reliable (see supporting information for AH sensitivity analysis)." (Page 5, lines 12-16).

(3) In the conclusion we added: "Use of AH changes during the winter and spring seasons was shown as a practical tool to identify the source of air parcels within the three geometries, namely atmospheric vs. lower-borehole/cavity, and thus to determine the direction and effect of the air transport." (Page 11, lines 15-17).

[Figure]

Figure. S2. Seasonal time series results of absolute humidity (AH) from the *large-diameter borehole*. From mid-February to the end of June, the use of AH as a proxy for airflow direction was valid due to the differences of AH between the borehole's lower boundary and the atmosphere. Between July and August (summer season), the AH differences were no longer constant (see also Fig. S3), and therefore we can conclude that AH was no longer a suitable proxy for the identification of airflow directions within the borehole. AH data for the borehole are missing from September until December; however, it is reasonable to assume that during that period, the AH as a proxy was also relevant because AH within the borehole was stable.

In Section 3.4, the authors contrast *winter* air movement detected in the large-D borehole with those noted in the shaft/borehole during *springtime*, when the external air has likely become far less dense. Convective instability for these "pipes" must generally be far greater for either geometry in February than in May. Nonetheless, the ensuing analysis suggests that dT/dz can be considered constant. There is no justification for this spurious assumption, particularly given that the authors have data that enable its direct assessment. I have little doubt regarding the authors' conclusion that convection is most important for the large-D borehole, whereas barometric changes dominate for the other geometries, but this needs to be demonstrated climatologically and without making invalid assumptions.

The purpose of this section was to demonstrate an ideal comparison in order to show the transition between governing air transport mechanisms with the increase of diameter (from BP to TIC). Following this comment and also the comment from the second reviewer regarding this analysis, we agree that indeed our comparison and assumptions were over- simplified. Therefore, we deleted the comparison paragraph that contained the potentially problematic assumptions. We left only the theoretical equations that can be used for a general discussion regarding the impact of the radius on both mechanisms (i.e., BP and TIC).

The paper would be greatly strengthened by broadening the dataset to include a seasonal range of temperatures/humidities for every geometry. Otherwise, the generalisations inferred by the authors should be reduced in scope to take into account their seasonal representativeness.

We agree. Unfortunately, due to technical restrictions and limitations, we could not obtain long-term measurements in some of our field sites. We broadened our *large-diameter borehole* dataset by a few months according to our available data. Because it is still not a full seasonal rang, we reduced our conclusion to a seasonal aspect as mentioned above in our answer to a previous comment: "…This mechanistic explanation was validated using the winter and spring season's dataset. Although we show that theoretically the transport mechanism observed for winter and spring should hold, with

reduced significance, for summer and autumn, further data are needed to verify the theoretical calculation" (Page 11, lines 26-28).

**Specific Comments**

In addition to the General comments above, which require significant revision to the analysis and presentation of the findings, I offer the following specific suggestions (by page/line) to improve the manuscript:

2/16 Do not neglect differences in air composition.

We added to the introduction the importance of RH and air composition to the overall density, including suitable references: "Although air density depends mainly on temperature, it is also depend on the air humidity, and to a lesser degree on the air's gas composition (Kowalski and Sánchez-Cañete, 2010). Integration of these three effects (temperature, relative humidity, and air composition) into a single parameter named virtual temperature ($Tv$) was proposed by Sánchez-Cañete et al. (2013). In environments of high $CO_2$ concentrations compared to the atmosphere, the importance of the gas composition on the $Tv$ becomes more pronounced. Such underground environments can be karstic areas of carbonate rocks (Sanchez-Cañete et al., 2011), caves (Denis et al., 2005; Guillon et al., 2015), and soils (Amundson and Davidson, 1990) where $CO_2$ concentrations can be very high, ranging from 10,000 to 100,000 ppm and above. For a given altitude level, the $Tv$ differences will determine the onset of TIC." (Page 2, lines 20-27).

3/11 Homogenize the font size "first 42 days"

Done.

3/13 That "the pipe touched the cavity floor" does not ensure a hermetic seal. Either the cavity floor geometry should be described (horizontal, smooth?), or the isolation of the pipe from the cavity should be somehow justified.

We validated this assumption by visual evidence from a camera that was lowered to the bottom part of the pipe – the pipe–soil interface. This is described in the materials and

methods section in the following sentence: "In addition, a televiewer was lowered into the pipe to verify that the pipe was intact and was either connected to or disconnected from the underground cavity in the shaft or borehole, respectively." (Page 4, lines 5-7).

4/1 In equations, it is preferable that each variable be represented by a single symbol, with necessary subscripts. A reader might interpret "AH" in equations (1) and also in the derivative at page 5, line 5 (and elsewhere) as the product of the variables "A" (volume flow rate per unit length) and H. Rather than "AH", I recommend the use of the standard rho (greek symbol for "r") with a subscript (v) to indicate that it is the vapour density that is synonymous with the absolute humidity. Similarly, relative humidity could be represented with "U" instead of "RH".

Done, AH in the equations was changed to $\rho_v$, $dAH/dt$ in the text was changed to $d\rho_v/dt$, and RH in the equations was changed to $U$.

5/16-19 The "typical" situation described is valid only for nighttime data (and is furthermore dependent upon season). Thus, a "decrease in temperature in the shaft observed by the temperature sensors" would not typically describe penetration during a daytime barometric increase since the external air temperature would exceed that of the cavity.

5/28-30 The ability to estimate the direction of the airflow in the shaft is not general. For example, a pressure change that occurred at a moment when the external air temperature coincided with the cavity temperature (as occurs twice per day in Fig. 2, and perhaps during many hours on summer nights) would not be detected.

Following the two comments above, we changed and combined these two paragraphs to a single paragraph. In the new paragraph, we added the above clarification that: "Stages 2 and 4 in Fig. 4 are valid mainly during winter and spring night-times when atmospheric temperatures are lower than within the borehole". (Page 6, lines 7-9). The lines describing the use of temperature sensors for airflow direction analysis were deleted.

6/24 Homogenize header format.

Done.

7/29 These cases that occur only at noontime are specific to winter, when the air temperature rarely exceeds the cavity temperature. In summer, it would be most often the case.

The sentence was changed to a more general definition which relates the air transport mechanism to thermal-stability and not to a certain time of the day: "The cases of AH increases at 10 m occurred when thermal-stability controlled the air movement inside the *large-diameter borehole*, and only then did AH at 10 m increase to the saturation conditions that characterize the lower boundary (marked as gray columns in Fig. 7a)" (Page 8, lines 7-9).

8/6-7 The validity of this hypothesis is very dependent on season. "Given the same climate conditions..." I agree, but the paper does not examine these geometries under the same climate conditions.

9/13 The assumption that dT/dz is constant makes no sense. Although BP and TIC are compared using the same climate conditions, TIC depends on dT/dz whereas BP does not. Increased values of dT/dz in winter make for greater convective instability, whatever the borehold diameter. But the geometries are not examined under the same climate conditions, since the shaft and borehole are examined in spring, whereas the large-D borehole is examined in winter.

We agree. These lines were changed as mentioned in the response to the major comments above.

8/11-etc. Much of the following 1.5 pages constitutes methodology, and would better be provided in section 2.

The equations presented in these pages are the basis for the discussion that follows them and therefore we think it is more suitable to present them in the scope of the discussion rather than in the methods section.

10/3 The conceptual model is a very good idea.

Thanks.

10/23-27 This paragraph regarding CO2 source/sink behavior is particularly weak, and I recommend deleting it entirely. To speak of sources/sink requires information regarding the origin/destiny of the CO2 in the cavity. In any event, I think it is absurd to state that the geometry is a source or sink. It might better be described as a storage space for CO2 that could otherwise have been emitted to the atmosphere (but from what source?). To appreciate the complexities of the source/sink issue, the authors are encouraged to consult section 3.1 of Serrano-Ortiz et al., 2010, Agric. and Forest Meteorol. 150, 321–329.

After reading the above reference, we realize that this paragraph is indeed too simplified, and we decided to delete it and change section 3.5 accordingly. We also added the above reference to direct the reader to the optional $CO_2$ sources in underground cavities (Page 10, Lines 28-29).

11/12 This "conclusion" is very much specific to site and season, and must be qualified as such or revised.

A clarification was added "Use of AH changes during the winter and spring seasons was shown as a practical tool to identify the source of air parcels within the three geometries, namely atmospheric vs. lower-borehole/cavity, and thus to determine the direction and effect of the air transport" (Page 11, Lines 15-17).

---

## Author Comment (AC2) · 15 Jul 2018

**Pipes to Earth's subsurface: The role of atmospheric conditions in controlling air transport through boreholes and shafts**

Letter of response:

Reviewer 2:

Earth's atmosphere is extremely complex system oneself and its interaction with nearsurface targets, deep dynamic geological-geophysical regularities, and some cosmic
factors (e.g., tidal effects) increases the total complexness.
Without hesitation, Levintal et al. have arisen very important problem of interaction between the underground caves, boreholes and mines with the Earth's atmosphere. This publication obviously will trigger a series of new publications in this field.
For instance, in the world a lot (tens of millions) of comparatively deep (> 500 m) boreholes were drilled in different physical-geological environments. Many of them are open, semi-open or have indirect connection with the Earth's atmosphere. Calculation of the total effect of air transport from these objects is a difficult physical-mathematical problem.

We would like to thank the anonymous reviewer for his comments and fruitful review. A detailed response for each comment is presented below in blue font.

**Some minor remarks are compiled below.**

I believe that 'boreholes' and 'mines' cannot be included to the class of 'caves' since they are principally different targets. Besides this, most part of caves are the natural geological objects existing sufficiently long time, whereas boreholes and mines are the artificial targets which have been appeared mainly in 20th century.

We agree that indeed caves can be quite different than boreholes, and we deleted the reference to caves from the conclusions section.

Generally speaking, examination of two (three ?) targets only is insufficient one. I propose that general conclusions done in this MS for all types of underground objects is untimely one.

The sentence that generalized our finding to other underground cavities was deleted.

CO2 concentrations in various underground targets strongly exceed the value of 2000 ppm (e.g., Guillon et al., 2015).

This was also commented on by Reviewer 1, and thus we also added a suitable clarification for environments with elevated $CO_2$ concentrations in the introduction: "In environments of high $CO_2$ concentrations compared to the atmosphere, the importance of the gas composition on the *Tv* becomes more pronounced. Such underground environments can be karstic areas of carbonate rocks (Sanchez-Cañete et al., 2011), caves (Denis et al., 2005; Guillon et al., 2015), and soils (Amundson and Davidson, 1990) where $CO_2$ concentrations can be very high, ranging from 10,000 to 100,000 ppm and above." (Page 2, lines 23-27).

I can suggest that the role of viscosity in air transport (Finkelstein et al., 2006) may be more significant than presented in the MS. The authors assumed some physical parameters as constant (for simplicity of calculations). It is a widely distributed approach and it is acceptable, for instance, for gravity acceleration and thermal expansion. However, accepting viscosity as constant is under question (Finkelstein et al., 2006). From numerous thermal measurements in wells follows that the behavior of dT/dz is not constant one (e.g., Huang et al., 2000; Eppelbaum et al., 2014). It should be taken into account in the further extension of this approach.

The purpose of this section was to demonstrate an ideal comparison in order to show the transition between controlling mechanisms with the increase of diameter (from BP to TIC). Following this comment, as well as the comment from Reviewer 1 regarding this analysis, we agree that indeed our comparison and assumptions were oversimplified (mainly that *dT/dz* and viscosity cannot be taken as constant). Therefore, we deleted the comparison paragraph that contained the invalid assumptions and left only the theoretical equations that can be used for a general discussion regarding the impact of the radius on both mechanisms (i.e., BP and TIC).

Obviously, an interaction between the near-surface targets and Earth's atmosphere has nonlinear character (e.g., Kardashov et al., 2000). It cannot be realized in the presented study, but can be reflected in future investigations.

We added to the discussion a clarification regarding this issue (section 3.4):"Moreover, some of the parameters presented in Eqs. (2) to (10) can exhibit nonlinear behavior (Kardashov et al., 2000), mainly $dT/dz$, which suggests that a comparison between sites is highly complex." (Page 10, lines 8-11). We thank the reviewer for the suggestion to implement this issue in further research, and we will take this into consideration in our next study.

I propose that after a small revision, this MS may be accepted for publication.

---

## Author Comment (AC3) · 15 Jul 2018

**Pipes to Earth's subsurface: The role of atmospheric conditions in controlling air transport through boreholes and shafts**

Letter of response:

Reviewer 3:

> This paper, after a noteworthy revision, could be a useful contribution to the literature on air transport and borehole-atmosphere exchanges. This manuscript studies the air transport between three borehole types using temperature and pressure gradients and using the water vapor as tracer of air flow. Levintal et al. found that the main mechanism driven the air transport depend on the geometries, where changes in pressure induce more transport in narrow boreholes and temperature differences induce more transport in wide-diameter boreholes.

We would like to thank the anonymous reviewer for his comments and fruitful review. A detailed response for each comment is presented below in blue font.

> My major comment to this paper is that the authors are studying the air transport neglecting the air composition to estimate the air mass buoyancy. The virtual temperature (Tv) is the temperature at which dry air would have the same density as the moist air, at a given pressure. In other words, two air samples with the same Tv have the same density, regardless of their actual temperature or relative humidity. Levintal et al are measuring the relative humidity in the external and internal air, therefore they would be estimate the virtual temperature to study the buoyancy.
>
> The authors mention that they found around 2000 ppm of CO2, however they don't show the CO2 pattern in any graph and the CO2 sensors are not described in methodology. I suggest showing the data as supporting information and discuss the possible influence in the air composition and its buoyancy.

We fully agree with both of the comments above, and also mentioned by Reviewer 1 (Prof. Kowalski). We revised our MS accordingly:

(1)  A summary of the virtual temperature concept ($Tv$) was added to the introduction: "Although air density depends mainly on temperature, it is also depend on the air humidity, and to a lesser degree on the air's gas composition (Kowalski and Sánchez-Cañete, 2010). Integration of these three effects (temperature, relative humidity, and air composition) into a single parameter named virtual temperature ($Tv$) was proposed by Sánchez-Cañete et al. (2013)…. For a given altitude level, the $Tv$ differences will determine the onset of TIC." (Page 2, lines 20-27).

(2)  In figures 2, 6, and 7, the data are now presented as a function of $Tv$ rather than T, according to Sánchez-Cañete et al. (2013).

We note that due to the fact that we didn't have continuous $CO_2$ measurements in this study, the $Tv$ within the boreholes was calculated using the T and RH continuous measurements with a constant $CO_2$ concentration according to preliminary results that we obtained in a two-week period during the 2017 winter season (~2000 ppm) ($CO_2$ sensor type – IRGAs GMD-20, Vaisala, Finland). Obviously, the $CO_2$ concentration varies throughout the day and seasonally within the boreholes; however, we think that this is a reasonable choice because we know from these two weeks of measurements that the $CO_2$ inside the borehole did not increase above 2000 ppm, and the values are relatively low compared to other underground cavities such as caves (e.g., Denis et al., 2005; Guillon et al., 2015). In addition, the $CO_2$ changes are only the third parameter in importance after temperature and RH for air density convection. Therefore, we believe that this is a reasonable assumption.

We also decided to delete the paragraph dealing with $CO_2$ emission in section 3.5 due to the scant $CO_2$ measurements that were available for us.

Minor comments (line/page):

1-5/3: How far are both sites? Could you include coordinates?

The distance between the sites was ~60 km. We added this information to the Materials and Methods section: "The distance between these sites is ~60 km." (Page 3, line 12).

11/3: Edit "42".

Done.

30/3: It's not clear how many sensors do you have and their positions? Please add more information and include this information in Fig 1b.

We added a sentence to the figure caption that describes the sensor location: "The sensor's location within the *shaft* included four thermocouples at depths of: 0, 6, 18, and 24 m and two RH-temperature sensors at the lower part of the *shaft* at its connection point to the cavity (27 m depth)." (Page 17, Lines 3-4).

8/4: delete "An example of"

Done.

13/4: the number 6050 will be 6048 (6 measurements/h*24h/day*42day)

Thank you, we missed calculated this number, and we changed it to 6048.

14-15/4: provide the % of relative humidity for 12 and 27 separately. In figure 1 and figure S1 I can see that during the whole period the relative humidity is always higher at 12m than at 27m, does that make sense?

We agree that this is not as expected. The differences are on the scale of a few percentages, which according to the manufacturer are close to the accuracy limit of the sensor (~±1 %). Also, from previous studies, we know that the sensors can drift a few percentages when they are under high RH for long periods of time. Therefore, to be more on the "safe side", we changed the comparison to values of RH> 90% and from 95%. The sentence was changed accordingly to: "From the overall 6048 measurements (42 days) at 12 and 27 m, 79 and 76 % of the RH values were above 90 %, respectively. The remainder of the measurements were no lower than a minimum of ~50 % RH (Fig. S1 – supporting information)" (Page 4, Lines 22-24).

9/7: Could it be because the max-min variation in temperature is higher in the shaft than in the borehole?

We agree, and this can further indicate that the barometric pumping effect on the air transport is greater in the shaft compared to the borehole; therefore, the max–min temperatures within the shaft are higher than the borehole (i.e., the atmospheric daily temperature fluctuations of air has greater effects on the shaft air than the borehole air).

24/7: could you provide some analysis (as a simple R^2) to prove that the correlation AH-Temperature is higher than AH-pressure?

While preparing this manuscript, we tried different statistical analyses to compare between both correlations. We found no compatible $R^2$ analysis that could be implemented with our results. We are now working on a new study in this field in which we are developing the use of more complicated statistical tools, mainly Fast Fourier Transform and Cross-correlation to correlate between atmospheric forces and airflow within boreholes.

> 3-5/8: I can see in Figure 7 that negative values on dPatm/dt increase the RH at 10m. During the days 22-24 probably the atmospheric pressure was changing from low to high pressure and for this reason the negative values on dPatm/dt were much lower. Would be useful if you show the atmospheric pressure value in a second Y-axis in Fig 7, panel 4.

Following a comment from Reviewer 1, the time series of this figure was changed to April 2017; thus, we can compare between similar seasons at both locations (spring season). However, we found this comment useful regardless of the specific day, and therefore we decided to add the Patm as an additional y-axis in panel 4 here (Fig. 7) and also in Fig. 2.

> 5-end/11: In your conclusions, you would remark that these conclusions were carried out only with data during 42 days in the shaft, 4 days in the borehole and 7 days in the large-diameter borehole, therefore, in the future we need investigate during longer periods,... :because for example in the case of the CO2, you found 2000 ppm in spring, but commonly the maximum values of CO2 are reached in summer/fall and therefore they could affect to the buoyancy.

The problematic generalization of our conclusions was also commented on by the other reviewers. We extended the large-diameter borehole data to the spring season, but unfortunately, we still don't have complete one-year results of the boreholes due to technical restrictions. We agree that this is a limitation in our study. We combined the comments from each of the reviewers and added this limitation in the conclusions section: "…This mechanistic explanation was validated using the winter and spring season's dataset. Although we show that theoretically the transport mechanism observed

for winter and spring should hold, with reduced significance, for summer and autumn, further data are needed to verify the theoretical calculation." (Page 11, lines 26-28).

Figure 2: In the legend "cavity air" is the Lower boundary, isn't it?

Yes, we changed it to "lower boundary" such that the definition will be consistent in all the figures. Thank you.

2-3/17: delete "representative"

Done.

4/17: change from "black dashed line in 3" to "…in panel 3"

Done.

Figure 7: increase the scale on panel 1 because the air temperature during the day 25 is chopped, and also move the legend box.

As mentioned above, the time series of this figure was changed. In any case, the legend box was moved so it will not block the upper y-axis.

---

## Author Comment (AC6) · 15 Jul 2018

The comment was uploaded in the form of a supplement:
https://www.earth-syst-dynam-discuss.net/esd-2018-18/esd-2018-18-AC6-supplement.pdf

---

## Author Response (AR2)

**Pipes to Earth's subsurface: The role of atmospheric conditions in controlling air transport through boreholes and shafts**

Letter of response:

Reviewer 1:

I believe that, following the revisions implemented by the authors, the manuscript should be accepted for publication. Nevertheless, in case they are of help when finalizing the paper, I offer the following suggestions (by page/line numbers in the revised manuscript).

We would like to thank Prof. Kowalski for his additional comments. A detailed response for each comment is presented below in blue font.

**Minor comments**

1/13  I would change "3.4 m and 59 m, respectively" to "59 m and 3.4 m, respectively", such that the order of depth and diameter is consistent in this sentence.

Done.

1/19: Change "$CO_2$ concentrations (~2000 ppm) were found" to "$CO_2$ concentrations of ~2000 ppm were found"

Done.

2/9: I remain unconvinced regarding the terminology "thermal-induced convection (TIC)". The raison d'être of the virtual temperature is that the density gradients that cause convection are not exclusively thermal, but can also be due to differences in air composition. Therefore, I suggest getting rid of the TIC terminology throughout the manuscript, and replacing it with something more appropriate. Perhaps "convective overturning (CO)" would work, since it is grammatically consistent with "barometric pumping (BP)".

TIC was deleted from the text. Using convective overturning (CO) can potentially confuse the reader to think CO is the abbreviation of Carbone monoxide gas so instead we used density-induced convection (DIC), which we think is relevant because the convection is driven by density gradients.

2/28: As it is written, this sentence seems to suggest a lower limit of 10,000 ppm for soil CO2 concentrations, which is certainly not correct. I suggest changing "ranging from 10,000 to 100,000 ppm and above" to "ranging as high as 100,000 ppm", or something similar.

The sentence was changed to: "…ranging as high as 100,000 ppm…" (2/27).

3/16: The negative sign (-80 m) is unnecessary since the sentence says "below the ground".

The negative sign was deleted.

3/30: I find it awkward that the authors have agreed to use "U" in equations to represent the relative humidity, but continue to refer to it as "RH" in the text. I would suggest simply defining the relative humidity (U) and consistently referring to it using this symbol.

We appreciate the comment and agree that both options are possible, with some pros and cons. Yet we believe that RH is a more common abbreviation in the relevant literature. Additionally, we are also using the RH terminology in our recent published manuscripts. Therefore, we prefer to stay with RH instead of U, convinced that it will be easier to follow for most potential readers.

4/9: Similarly, I would change this sentence to say "The absolute humidity, or water vapor density (rho_v) was used...", and then to use rho_v instead of AH throughout.

Again, we do understand that there are two options here, but prefer to stay with the existing terminology for the same reasons explained above.

6/32: "when negative pressure changes occur"

Done.

7/21: change "two-" to "two"

Done.

8/7: Change "thermal-stability" to "stratification"

Done.

9/3: In mathematics, two variables are proportional if there is always a constant ratio between them. The constant is called the coefficient of proportionality or proportionality constant. Because the exponent relating the variables u and r is 1.85, I

think it is incorrect to say that they are (inversely) proportional. The same applies to the sentence at page 9 line 15.

In both lines, the word "proportional" was changed to "correlated" to avoid this mathematical error.

11/1: change "Methane, which is emissions" to "Methane, whose emissions"

Done.

12/10: Kowalski is misspelled.

We apologies, the name is now spelled as "Kowalski".

[revised manuscript text omitted]